# The Relationship between Audit Adjustments and Audit Quality in Iraq

Mahdi Salehi [1,*], Mohammed Ibrahim Jebur [1], Saleh Orfizadeh [2] and Ali Mohammed Abbas Aljahnabi [1]

1    Department of Economics and Administrative Sciences, Ferdowsi University of Mashhad, Mashhad 9177948974, Iran; mohameedibraheem64@gmail.com (M.I.J.); aljanabiali801@gmail.com (A.M.A.A.)
2    Department of Accounting, Shahrood Branch, Islamic Azad University, Shahrood 3614773955, Iran; saleh.orfizadeh2010@gmail.com
*    Correspondence: mehdi.salehi@um.ac.ir

**Abstract:** The present research aims to assess the potential impact of audit adjustments on the audit quality for the listed companies on the Iraqi Stock Exchange. In other words, this research attempts to answer whether the audit adjustments can improve the audit quality or not. To achieve the objectives, a multivariate regression model was employed to test the hypotheses. The research hypotheses were tested using a sample of 35 listed firms on the Iraqi Stock Exchange for 2014–2020 by exploiting a multiple regression model based on the panel data technique. The research findings indicate a positive and significant relationship between audit adjustments and quality. Such findings demonstrate that audit adjustment might be considered a quality factor for audit services. Since no research has addressed such a subject in Iraq, the study can provide helpful information for the equity owners, board of directors, and audit firms, contributing to developing science and knowledge in the auditing field of study.

**Keywords:** audit adjustments; audit quality

## 1. Introduction

The quality of audit services has long been the concern of the authorities, for which standard-setters around the world made remarkable adjustments in the audit reporting standards to enhance the transparency of the audited financial reports and generally the quality of reporting process. For example, the Public Company Accounting Oversight Board in the US in 2017 implemented a new auditor reporting standard, requiring the auditors to describe the key audit matters in the audit report (PCAOB 2017), as well as the Financial Reporting Council (FRC) in the UK setting a new standard in 2013 requiring audit firms to disclose risks of material misstatement in their audit reports (FRC 2008). Both of these imposed requirements show the importance of audit reports' quality, particularly in cases that may exacerbate the risk of misstatements in financial reports. To be more precise, PCAOB's Final Rule in the 2019 Auditor's Report argues that "the communication of critical audit matters could further incentivize auditors to demonstrate the level of professional skepticism necessary for high quality audits in the areas of critical audit matters" (PCAOB 2017, p. 81). Empirical studies also reveal the prominence of audit quality in financial reporting systems. In this regard, in a comprehensive meta-analysis, Salehi et al. (2019b) articulate that the audit firm size, audit specialization, and audit tenure are more likely to improve the quality of audit services. Moreover, in another effort, Salehi et al. (2019a) show that factors such as audit fee stickiness and financial reporting quality are less likely to determine the level of audit quality. In addition, scholars document that audit quality may have a significant impact on other factors such as audit fees (Daemigah 2020; Cho et al. 2021), firm performance (Al-ahdal and Hashim 2021), and corporate governance mechanisms (Firnanti and Pirzada 2019).

Alternatively, it is highly expected that the auditors should prevent unrealistic financial statements about firm managers since they make the most important investment decisions



based on the auditor's report and opinion on these financial statements. Therefore, discussing the auditing adjustments and the significant misstatements in financial statements is particularly important. Thus, the present study attempts to analyze the phenomenon's effect on the audit quality of financial statements because the population and users seek auditors with the highest possible quality to rely on his/her report to make the most critical investment decisions. Audit quality is at the heart of the International Auditing and Assurance Standards Board (IAASB). Therefore, to provide the auditors' observation, the board considers it necessary to increase the audit quality in line with the public interests and states that "the board's purpose is to focus on professional complaints and control the audit quality". In this regard, the board's published audit quality framework in 2014 considers the auditors and auditing firms' role and other stakeholders' audit quality. It explains the content factors that have a significant impact. This framework describes the inputs, processes, and outputs that affect the audit quality at the level of business units, audit firms, and internationally that affect the audited financial statements. This framework also shows the importance of the appropriate interaction among stakeholders and the importance of the different factors (IAASB 2014). The audit quality and the determining factors have become important in the academic and supervisory discussions on auditing after the reported failures in this profession in recent years. As a result, many auditing profession developments, financial reporting, and the rules and standards' regulatory boards improve the auditor's quality. One of these regulators is the International Auditing and Assurance Standards Board (IAASB), which published a framework for audit quality in 2014 discussing the related factors to audit quality in practice. In 2008, the Financial Reporting Council published an article on the auditor quality's objectives and skeleton, stating the most critical aspects of audit quality. Similarly, the Institute of Chartered Accountants' England (ICAEW 2008) published a report on audit performance concepts and factors. Different researchers (such as Francis 2011; Knechel et al. 2013; Simnett et al. 2016) have examined audit quality firms in academic fields. Because the users make decisions about the retention or selling of their investments based on the auditor's comment on the financial statements, considering the importance of audit quality, the present research investigates the impact of audit adjustments on the quality of services provided by Iraqi audit firms. In other words, the present study seeks to answer the question of whether these audit adjustments can affect the audit quality or not.

Audit adjustments apply when the auditor asks the firms to provide earnings before the year-end audit; thus, it is incorporated with the quality of reported earnings and other figures contained in financial statements. The financial accounting and auditing literature criteria for measuring audit adjustments include earnings volatility, earnings smoothing, accruals quality, earnings liquidity, and combined index of accounting earnings (Etemadi et al. 2009). Auditing financial statements is an essential tool of the financial markets that provide reliable and valuable information about the financial statements' decisions by increasing the credibility and reliability of financial statements. It is also considered one of the most important goals of financial reporting. In the meantime, the phenomenon of annual adjustments has been of interest to researchers in recent decades. Understanding the auditing services' market status and competition level in the auditing profession is essential. DeAngelo (1981) states that auditing will have the quality to identify, evaluate, and report these significant audit adjustments. The first dimension of this definition refers to the professional competence, the degree of expertise, and the knowledge that the independent auditor benefits. The next dimension also refers to the level of his/her independence. Therefore, audit adjustments can represent the audit quality degree because these adjustments include material, minor, and fundamental misstatements, leading to the restatement of the financial statements. Suppose these misstatements are not identified and reported, and the auditor adjusted his report accordingly and submitted the correct report. In that case, the opinion will lead to the stakeholders' wrong decision, according to the auditing and accounting standards and even based on the agency theory (Jensen and Meckling 1976), the informational and signaling asymmetry.

The present study is among the pioneering attempts to explore the impact of audit adjustment on the quality of audit reports, particularly in an emerging market such as Iraq. Such a claim is made according to several channels; (1) prior investigations have mostly emphasized on the role of audit efforts (Xiao et al. 2020), cross-year updates of analysts' EPS forecasts (Liu and Chen 2019), client importance (Chen et al. 2018), and auditor's communication mode (Saiewitz and Kida 2018) on the level of audit adjustment; (2) previously conducted studies have mostly revealed that auditors related indicators such as audit specialization, tenure, and audit firm size (Salehi et al. 2019b), audit fee stickiness (Salehi et al. 2019a), audit market concentration (Gunn et al. 2019), and client-related features such as board size, client firm size (Akinyomi and Joshua 2022), technology enhancement client firm size (Yanti and Wijaya 2020), board diversity, and audit committee characteristics (Mustafa et al. 2018); and (3) to the best of the authors' knowledge, the closest conducted paper to this study might be the study of Lennox and Wu (2022) exploring how audit adjustments may link the effect of mandatory internal control audits on financial reporting quality, in which there are remarkable differences between these two. For example, Lennox and Wu (2022) estimate the moderating role of audit adjustment on the association between internal control mechanisms and misstatements in clients' pre-audit financial statements. In contrast, the objective of this study is to explore the direct impact of audit adjustment on the quality of auditors' reports. Drawing inference, the current investigation will likely add to the auditing and financial reporting literature to some extent.

## 2. Literature Review

### 2.1. The Theoretical Issues and Hypotheses Development

One of the main concerns of the auditing firms, rules, provisions, regulators, and investors is understanding the audit quality, one of the fundamental issues in the accounting literature (Khudhair et al. 2019). Audit quality improves financial reporting by increasing the financial statements' accuracy and credibility (DeFond and Zhang 2014). Conceptually, the audit quality can be a function of the accounting processes (such as the accounting systems, the internal controls, the economic transactions, the regulations) and the working personnel in the audit firms and the business units of the customers to perform those processes (accountants, auditors, and managers) (Francis 2011). Hence, these individuals can play a crucial role in the audit process. The audit personnel's role is also of particular importance. One of the auditing profession' primary tasks is recruiting and training auditing personnel (Francis 2011). The expertise and the quality of auditing personnel can be increased because audit quality plays a significant role in strengthening confidence in the credibility and accuracy of the financial statements critical to improving business units (Farouk and Hassan 2014). The audit quality also ensures that the users of the financial statements can rely on the auditor's report when deciding on their investments (Elewa and El-Haddad 2019). Audit quality refers to the services the involved auditors provide in the business. The business units seek auditing with higher quality because of the standards and experience. Hiring high-quality auditors attract more investments, improves organizational performance, and reflects a favorable image of the business unit (Khudhair et al. 2019). Hence, stakeholders and investors trust and rely on auditing firms of higher quality because of the credibility and the experiences that higher quality-accounting firms have gained (Khudhair et al. 2019). To better understand the concept of audit quality, several studies (DeAngelo 1981; Francis 2004; Duff 2004; Francis 2011; DeFond and Zhang 2014; Ellwood and Garcia-Lacalle 2015; Elewa and El-Haddad 2019; Farouk and Hassan 2014; Khudhair et al. 2019) have been conducted and different definitions have been given as follows: DeAngelo (1981) proposed the first definition of audit quality as "assessing the market for the auditor's ability to detect significant distortions and report detected distortions". DeAngelo (1981) also emphasizes that the auditor will be independent of the word's true sense by detecting and reporting inaccuracies and distortions. Thus, according to DeAngelo (1981), audit quality increases the audit's ability to detect accounting distortions and evaluate the population's ability and independence. When DeAngelo proposed this

definition of quality, the basic assumption was that society understands that audit quality reflects the true quality of the auditing (Moizer 1997). Palmrose (1988) defined audit quality as the assurance of the financial statements and the likelihood that the financial statements will not have any important misstatement. In practice, the definition is used due to the audit process. On the other hand, audit adjustment is used to amend the auditor's financial statements to the management's management and governing skeleton (Greenwood and Zhan 2019). The auditor makes this amendment based on the obtained evidence during the audit stages, leading to the re-submission of the financial statements or even claims from the management to classify the amount for the different accounts' financial statements. These adjustments should have a relatively high financial burden because the owner may refuse to correct them (Lennox et al. 2016). The auditing research shows that audit quality is closely associated with the auditing industry. Audit quality increases by increasing expertise in the audit industry (McLelland and Giroux 2000; Deis and Giroux 1992). Other research studies have also shown that audit quality decreases with the clients' size and financial wealth (McLelland and Giroux 2000; Deis and Giroux 1992). Some have also shown that the auditor's size and credibility do not affect the audit quality (Copley 1991; Ballantine et al. 2008). Lennox et al. (2016) examine the impact of the audit adjustments on the audit quality and the earning quality in Chinese firms. His research indicated that the audit adjustments have a positive and significant relationship with audit quality and earning quality and reduce earnings management. Lennox et al. (2018) also showed that audit adjustments increase the earning quality and enhance financial reporting quality. The research results show that the managers are less inclined to audit adjustments (Greenwood and Tao 2017). According to the agency representation, auditing reduces agency costs (Jensen and Meckling 1976; Watts and Zimmerman 1983) and reduces the possibility of automated reporting by the management. Research (e.g., Hoerger 1991; Leone and Van Horn 2005; Ballantine et al. 2008; Greenwood and Tao 2017) shows that management reporting higher revenues and lower expenditures, which has been particularly prevalent since accrual accounting, (Barton 2009), increases the accrual earnings management opportunities (Hood 1991, 1995; Lapsley 2009). Therefore, if the auditors reduce the agency costs, we expect them to reduce the accrual earnings management through the audit adjustments. However, the auditors are representatives in their own right who will seek to identify and report the earnings management to gain a reputation as a stakeholder to avoid litigation and maintain their reputation (DeAngelo 1981; Antle 1982, 1984; Francis 2004; Francis 2011). Therefore, to not lose the reputation and file lawsuits against the auditors, the auditors perform a higher quality audit process. In other words, the legal claims and the risk of losing the reputation are the motivating factors to improve the audit quality of the auditors in the audit process (Francis 2004, 2011; DeFond and Zhang 2014; Francis and Wilson 1988). The discussions mentioned above present the underlying theoretical frameworks for hypothesis development in the following section.

### 2.2. The Relationship between the Audit Adjustments and the Audit Quality

According to agency theory, auditing is a tool to reduce agency costs (Jensen and Meckling 1976; Watts and Zimmerman 1983) and prevent any unrealistic reporting by the management of all the private, for-profit, and non-profit departments and units. Since there is a weaker incentive framework in the public than in the private sector, when the surplus account of income declines, it is close to zero (Hoerger 1991; Leone and Van Horn 2005; Ballantine et al. 2008; Greenwood and Tao 2017). That has been formed since the advent of commitment accounting (Barton 2009), which has created new earnings management opportunities (Hood 1991, 1995; Lapsley 2009). In this field, much evidence shows that earnings management with accruals or so-called accrual earnings management has occurred more in government units than in the private sector and is more aggressive (Vermeer et al. 2014). Therefore, if the auditor seeks to reduce the agency costs, we expect him/her to negotiate with the management to adjust these items in the financial statements. It is worth mentioning that auditors also pursue their interests and seek to reduce lawsuits against



themselves and prevent losing their credibility (DeAngelo 1981; Antle 1982, 1984; Francis 2004, 2011). Therefore, as a supervisory agent, the auditor may act in his favor rather than following the principles and criteria. On the other hand, some also believe auditors can prevent applying business unit managers' earnings management through audit adjustments. Furthermore, the business management may not accept the audit adjustment, especially if these adjustments reduce the remuneration given to the business management at the end of the year or even reduce the credit finance borrowing. Therefore, in these situations, the auditor should consider whether these audit adjustments have a significant impact on the accuracy and correctness of the financial statements or not or whether these adjustments, in turn, can lead to the fact that the auditor can make a correct and acceptable opinion about the financial statements of that unit or not (Lennox et al. 2018). Of course, in most cases, the owner accepts the audit adjustments because he seeks to receive the auditor's opinion about himself/herself and a favorable opinion of him/her. If the business units have an audit committee, auditors of the financial statements negotiate the adjustments with them (Greenwood and Zhan 2019). By examining the possible problems in the controls or other cases, the committee members also investigate the accuracy of transaction recordings of the accounting unit to determine the unit's effectiveness. The process can lead to a change in the management of the entity's accounting department. The final discussion about the audit adjustments is to review the initial account balances at the beginning of next year to ensure that the owner has correctly recorded all the audit adjustments. If not recorded, these adjustments will affect his/her report and lead to an auditor's adjusted comment (Lennox et al. 2016, 2018). According to DeAngelo's (1981) definition of audit quality, an audit will be effective and high-quality to identify the financial statements' significant misstatements and report these distortions without financial or non-financial dependence. In other words, this definition's first and second parts refer to the auditor's expertise and independence. Having expertise about the affairs and providing an opinion contrary to the owner's expectation determines the audit quality. According to the definition, the auditor's ability to identify the significant distortions that lead to the adjustment of the financial statements depends on his/her degree of expertise and independence. Therefore, audit adjustments and quality are closely related (Lennox et al. 2018). Research done in different countries is as follows: Greenwood and Zhan (2019) showed a positive and significant relationship between audit adjustments and the public sector audit quality. Lennox et al. (2016, 2018) also stated that audit adjustments increase earning quality and improve business financial reporting quality. Moradi et al. (2020) showed no association between incremental profit audit adjustments and companies' financing. There is also no meaningful relationship between the earnings downward/upward audit adjustments and companies' financing. However, there is a significant relationship between the profit downward audit adjustments and firms' financing through a loan. Lennox et al. (2016) suggest that the annual adjustments increase the auditors' conservatism and prevent the business unit managers from making profits. Donatella (2021), investigating the extent to which audit firms mitigate management bias in public sector financial reports, finds that auditors act to reverse management bias in the case of trusts with a pre-audit deficit but finds no evidence in the case of trusts with a pre-audit surplus. The results of Xiao et al. (2020) show that audit effort significantly increases the probability of audit adjustments, which inhibits positive earnings management and improves the quality of audited financial statements. They also find that audit effort does not significantly affect the issuance of modified audit opinions overall but that a modified audit opinion is more likely to be issued in the absence of an audit adjustment. Furthermore, they find that the impact of audit effort on audit quality is attenuated when clients are more complex and when audit firms are larger. Collectively, their evidence suggests that audit effort is important in improving audit quality by influencing audit process and output. Bronson et al. (2021) find that audits that are less complete at the earnings announcement date are associated with a higher likelihood of financial statement misstatements in audit areas typically performed toward the end of audit fieldwork. We also find a higher likelihood of auditor turnover

during the following year. The results suggest lower financial reporting/audit quality and higher auditor turnover for companies that release earnings when the audit is less complete. Lennox and Wu (2022), examining whether audit adjustments are a mechanism that links the effect of mandatory internal control audits on financial reporting quality, argue that the requirement for auditors to disclose internal control weaknesses publicly exacerbated auditor–client conflicts and that this resulted in auditors being less likely to detect (and correct) misstatements in their clients' pre-audit financial statements. Consistent with such an argument, they find significant reductions in audit adjustments following the staggered introduction of mandatory internal control audits. They also find that reductions in audit adjustments are associated with significant increases in material misstatements following mandatory internal control audits. In contrast, they find that the introduction of mandatory internal control audits led to a significant reduction in material misstatements among clients that did not experience reductions in audit adjustments. Overall, the two effects offset each other, which explains why financial reporting quality did not improve, on average, following the introduction of mandatory internal control audits. However, Choudhary et al. (2022) find that waived adjustments are linked to lower financial reporting quality measured by material misstatements, to incentives to meet/beat earnings targets, and to the audit process, as measured by higher next-period audit effort and fees and higher next-period proposed adjustments. These effects on the audit process are consistent with auditors responding to the increased risk associated with waived adjustments. In an exploratory analysis, they find that controlling for the number of proposed adjustments, auditor resignations are negatively associated with waived adjustments.

Therefore, we investigate and prove the present study's relationships with the Iraqi Stock Exchange firms. Thus, according to what was said, the research hypothesis is as follows:

**H1.** *There is a significant relationship between audit adjustments and audit quality.*

## 3. Research Methodology

The present study's statistical population was all listed firms on the Iraqi Stock Exchange between 2014 and 2020. The systematic elimination method was used for sampling, and finally, after applying the following conditions, the statistical sample of the research was selected:

1. They are accepted on the Iraq Stock Exchange by the end of 2014.
2. They provide the required financial information to complete this research during the research period.
3. They should not be affiliated with investment companies, banks, insurance, or financial intermediaries.

According to the collected information at the end of 2020, the final sample was obtained based on Table 1.

### 3.1. The Method and Data Collection

The required raw information and data to test the hypotheses using the Iraqi Stock Exchange database and the Iraqi Stock Exchange's published reports through a direct reference are presented as a CD by the Iraqi Stock Exchange Organization.

### 3.2. The Method of Data Analysis

The method of data investigation was cross-sectional and year-to-year (data panel). The multivariate linear regression method was used to test the hypotheses, and descriptive and inferential statistical methods were applied to analyze the obtained data. Thus, the frequency distribution table was used to describe the data. At the inferential level, the F-Leimer test, the Hausman test, the normality test, and the multiple linear regression test were used to test the research hypotheses. Furthermore, the R studio statistical software analyzed the raw data.

**Table 1.** The number of firms included in the statistical population by imposing the conditions for Iraqi sample selection.

| Listed Firms on the Iraqi Stock Exchange | Firm No. | Eliminated Firms | Selected Firms |
|---|---|---|---|
| Bank firms | 39 | 39 | |
| Insurance firms | 5 | 5 | |
| Investment firms | 9 | 9 | |
| Service firms | 10 | 4 | 6 |
| Industrial firms | 25 | 10 | 15 |
| Hotel and tourism firms | 10 | 2 | 8 |
| Agricultural firms | 6 | 0 | 6 |
| Telecommunication firms | 2 | 2 | |
| Financial delivery firms | 17 | 17 | |
| Total sample firms | 123 | 88 | 35 |

*3.3. The Research Variables*

3.3.1. The Dependent Variables

AQ: The audit quality is equal to the obtained accruals quality from the following relationship:

The Jones-adjusted model (1991) and Dechow et al. (1995) are used to calculate discretionary accruals. First, coefficients (2) are estimated:

$$\frac{TA_{i,t}}{Assets_{i,t-1}} = \alpha_1 \left( \frac{1}{Assets_{i,t-1}} \right) + \alpha_2 \left( \frac{\Delta Sales_{i,t}}{Assets_{i,t-1}} \right) + \alpha_3 \left( \frac{PPE_{i,t}}{Assets_{i,t-1}} \right) + \varepsilon_{i,t} \quad (1)$$

After estimating the coefficients, the non-discretionary accruals are calculated using the third relationship.

$$\frac{NDA_{i,t}}{Assets_{i,t-1}} = \alpha_1 \left( \frac{1}{Assets_{i,t-1}} \right) + \alpha_2 \left( \frac{\Delta Sales_{i,t} - \Delta AR_{i,t}}{Assets_{i,t-1}} \right) + \alpha_3 \left( \frac{PPE_{i,t}}{Assets_{i,t-1}} \right) \quad (2)$$

Finally, we have to calculate the discretionary accruals:

$$\frac{DA_{i,t}}{Assets_{i,t-1}} = \frac{TA_{i,t}}{Assets_{i,t-1}} - \frac{NDA_{i,t}}{Assets_{i,t-1}} \quad (3)$$

In the above relationships, TA are accruals, Assets are Total Assets, Sales is the income, AR is the Accounts Receivable, PPE is property, plant, and equipment, NDA is the non-discretionary accruals, DA is the discretionary accruals. In this study, the following formula was used to calculate accruals items that are known as the profit and the loss:

*Operating Cash Flow − Income before the Extraordinary Items = Accruals*

Many previous studies used discretionary accruals (DA) to measure earning and audit quality. This study used the DA as a proxy for the audit quality because it provides a degree of the related negotiations to the audit adjustment decisions. The unusual performance adjustment accruals estimate the size of the DA.

3.3.2. The Independent Variables

AuditA: Audit Adjustments: The average variable of change in the company and the revenues in the year under review.

### 3.3.3. The Control Variables

AFA: The firm's audit age is equal to the time interval between the establishment of the firm and the year under review.

Size: The company's size is equal to the company's natural logarithm assets in the year under review.

ROA: Return on assets equals the net dividend divided by the company's total assets.

ROE: Return on equity equals the net dividend divided by the book value of equity.

Growthsales: The sales growth equals this year's sales minus last year's sales divided by last year's sales.

Age: The company's age is equal to the time interval between the date of the establishment and the year under review.

Atenure: The auditor's tenure and duration have consistently been under review in the entity's audit position.

Achange: The Auditor change if the auditor has changed in the year under review, number one; otherwise, number zero.

Rest: The restatement of the financial statements is a constant variable equal to the number one; if the financial statements are restated; otherwise, the number is zero.

Industry: The dummy variable of industry.

Year: The dummy variable of the year.

### 3.4. Research Model

The following multiple regression model was used to test the research hypotheses:

Model (1)

$$AQ_{it} = a_0 + a_1 AuditA_{it} + a_2 Atenure_{it} + a_3 Achange_{it} + a_4 Big1_{it} + a_5 size_{it}$$
$$+ a_6 ROE_{it} + a_7 ROA_{it} + a_8 growthsales_{it} + a_9 AFA_{it} + a_{10} Rest_{it}$$
$$+ a_{11} age_{it} + a_{12} Industry_{it} + a_{13} year_{it} + \varepsilon_{it}$$

## 4. The Data Analysis Results

### 4.1. The Descriptive Statistics

This study used one model to examine the relationship between audit adjustments and the audit quality of the Iraqi audit firms. The present study also included the panel data method of 34 firms in its database.

The variables of audit adjustments, audit quality, and control variables were used to estimate the model. Table 2 briefly shows the related information to the model variables.

**Table 2.** Descriptive statistics of research variables.

| Sign | No. of Observation | Mean | Std. Dv. | Min. | Max. |
| --- | --- | --- | --- | --- | --- |
| AQ | 264 | −0.232 | 0.682 | −4.976 | 0.999 |
| AuditA | 264 | 0.592 | 0.967 | −6.215 | 0.989 |
| Atenure | 264 | 3.325 | 2.043 | 1.000 | 9.000 |
| Achange | 264 | 0.193 | 0.396 | 0.000 | 1.000 |
| BIG1 | 264 | 0.632 | 0.483 | 0.000 | 1.000 |
| SIZE | 264 | 15.444 | 1.353 | 10.655 | 19.389 |
| ROE | 264 | 0.052 | 0.310 | −1.762 | 0.909 |
| ROA | 264 | −0.009 | 0.189 | −1.095 | 0.338 |
| Growthsales | 264 | 0.045 | 0.639 | −1.0002 | 2.901 |
| REST | 264 | 0.091 | 0.288 | 0.000 | 1.000 |
| Age | 264 | 32.239 | 14.066 | 9.000 | 71.000 |

On average, about 9% of the sample companies submitted their financial statements in the year under review. Moreover, about 63% of the sample institutions consist of large institutions. The average tenure period in Iraq is three and a half years. The average amount of audit adjustments per year has been 60%.

### 4.2. The Linearity

Table 3 shows the results of the linearity test of the research models:

**Table 3.** The results of the linearity test.

| Variable | VIF | 1/VIF |
|---|---|---|
| Atenure | 1.96 | 0.510 |
| Achange | 1.54 | 0.650 |
| BIG1 | 1.40 | 0.712 |
| ROA | 1.28 | 0.778 |
| AuditA | 1.21 | 0.823 |
| SIZE | 1.18 | 0.846 |
| ROE | 1.17 | 0.856 |
| Age | 1.16 | 0.862 |
| Growthsales | 1.04 | 0.963 |
| REST | 1.03 | 0.971 |
| Mean VIF | 1.30 | |

As shown in the table above, according to the obtained VIF statistic, which was less than 10 for all variables, there is no linearity among the model variables.

### 4.3. The Sensitivity Analysis

As shown in Table 4, the sensitivity analysis, examines the relationship between the model's used variables in pairs. The above matrix's output, whose diameter examines the correlation between the variable and itself, is always one, meaning complete correlation. If these numbers are closer to one, the correlation is more, and if the numbers are closer to zero, they will be without correlation. The correlation range is between negative 1 and positive 1. The negative numbers indicate an inverse correlation, and the positive numbers show a direct correlation.

**Table 4.** The results of the sensitivity test.

| | AQ | AuditA | Atenure | Achange | BIG1 | ROE | ROA | Age | Growthsales | REST | SIZE |
|---|---|---|---|---|---|---|---|---|---|---|---|
| AQ | 1.000 | | | | | | | | | | |
| AuditA | −0.174 | 1.000 | | | | | | | | | |
| Atenure | 0.033 | −0.019 | 1.000 | | | | | | | | |
| Achange | −0.053 | −0.134 | −0.558 | 1.000 | | | | | | | |
| BIG1 | −0.117 | 0.028 | −0.445 | 0.194 | 1.000 | | | | | | |
| ROE | −0.043 | 0.185 | 0.065 | −0.029 | −0.005 | 1.000 | | | | | |
| ROA | −0.210 | 0.090 | 0.106 | −0.036 | −0.211 | 0.315 | 1.000 | | | | |
| Age | 0.026 | −0.205 | −0.143 | 0.069 | 0.085 | −0.052 | −0.278 | 1.000 | | | |
| Growthsales | 0.024 | 0.015 | −0.029 | 0.043 | 0.036 | 0.115 | 0.147 | −0.004 | 1.000 | | |
| REST | 0.033 | 0.087 | −0.115 | 0.012 | 0.022 | 0.001 | 0.021 | −0.048 | −0.032 | 1.000 | |
| SIZE | −0.476 | 0.258 | 0.101 | −0.020 | 0.169 | 0.109 | 0.047 | −0.029 | 0.066 | 0.023 | 1.000 |

### 4.4. F-Limer Test

First, the data should be checked for pooled or panel by F-test to estimate the patterns. The null hypothesis in this test indicates pooled data, and hypothesis one indicates the panel data. Suppose the H0 is rejected after performing the F test. In that case, the question arises as to which of the model's fixed and random effects models can be examined, determined by the Hausman test. According to the presented integration test results in Table 5, H0 declares that data pooled at 99% confidence for the research model's data are rejected. Therefore, the panel data model for the research model should be used to estimate the model coefficients.

**Table 5.** The results of the pooled test.

|  | Probability Level | Calculated Statistic |
|---|---|---|
| Research model | 5.30 | 0.000 * |

* Significance level at 99%.

### 4.5. Hausman Test

Table 6 shows that the Hausman test statistic based on the estimate for the research model is 81.49. Therefore, according to Table 6, H0 is not rejected. Thus, the appropriate model for the research model is a model with random effects.

**Table 6.** The results of the Hausman test based on data.

|  | Probability Level | Calculated Statistic |
|---|---|---|
| Research model | 81.49 | 0.000 * |

* Significance level at 99%.

### 4.6. The Estimation Results of the Research Model

Table 7 show a positive and significant relationship between audit adjustments and quality because its *p*-value is 0.000, less than the significance level of 0.05. Its coefficients are also positive number 0.133. This indicates that the study's hypothesis is accepted, which states a positive and significant relationship between the audit adjustments and quality. This means the corrections of financial statements proposed by Iraqi auditors may significantly improve the audit quality, resulting in further transparency and the accuracy of accounting figures in financial statements. For example, clients probably receive greater adjustments for their financial reports and are less likely to insert manipulations, including earning management or smoothing, while preparing the accounting numbers for reporting purposes (Ismael and Kamel 2021). In this regard, Le and Moore (2021) argue that the higher audit quality may experience lower income-increasing discretionary accruals. Additionally, the statistics show that on average, 13 percent of the quality of audit services provided by auditors in Iraqi is likely to be explained by their proposed adjustments.

Table 7 shows the Robust model's estimation results (1). The panel data model examines four classical econometric assumptions and reports reliable results. These four assumptions include linearity among the variables, extrapolation of the explanatory variables, a variance of homogeneity, and the lack of serial auto-correlation among the disturbing components.

According to the applied regressions for companies, the origin's width is meaningful, but the significance level of the model is 0.000, which is less than the significance level of 5%. Hence, the model is of good and sufficient significance.

**Table 7.** The results of model (1) estimation.

| Variable (AQ) | Coef | Std. Err. | z | Prob |
|---|---|---|---|---|
| AuditA | 0.133 | 0.027 | 4.84 | 0.000 *** |
| Atenure | −0.0003 | 0.015 | −0.02 | 0.982 |
| Achange | 0.011 | 0.066 | 0.17 | 0.868 |
| BIG1 | −0.066 | 0.099 | −0.67 | 0.503 |
| SIZE | −0.010 | 0.024 | −0.45 | 0.656 |
| ROE | 0.059 | 0.076 | 0.79 | 0.432 |
| ROA | −0.320 | 0.163 | −1.96 | 0.050 ** |
| Growthsales | 0.036 | 0.035 | 1.04 | 0.297 |
| REST | 0.056 | 0.074 | 0.77 | 0.443 |
| Age | 0.001 | 0.003 | 0.38 | 0.704 |
| _con | 2.703 | 0.419 | 6.44 | 0.000 *** |
| Obs | 264 | | | |
| R-SQ | 0.4540 | | | |
| R-SQ2 | 0.2733 | | | |
| Prob Model | Wald chi2 (10) = 32.46 | | | |
| | Prob > chi2 = 0.0003 *** | | | |

*** Significance level at 99% and ** Significance level at 95%.

### 4.7. The Robustness Test

#### 4.7.1. The Results of the (Research) Model Estimation Based on the Ordinary Least Squares (OLS) Method

To ensure that the research results are robust, we tested the research model with different methods to determine whether the results aligned with the main method's results. According to the above Table 8 results, based on the ordinary least squares method, there is a positive and significant relationship between audit adjustments and quality because its $p$-value is 0.000. This is less than the significance level of 0.05. Furthermore, its coefficients are also a positive number of 0.162, which indicates that H0 has a positive and significant relationship with audit adjustments and quality, which is completely consistent with the main research method and confirms that.

#### 4.7.2. The Results of the Research Model Estimation Based on the Arellano–Bover Model Generalized Method of Moments (GMM)

To ensure that the research results are robust, we tested the research model with different methods to determine whether the results aligned with the main method's results. According to the above Table 9 results, based on the Arellano–Bover model's Generalized method of moments (GMM), there is a positive and significant relationship between the audit adjustments and quality. Furthermore, because its $p$-value is 0.000 is less than the significance level of 0.05, and its coefficients are also the positive number of 0.172, the main research method's results are confirmed.

**Table 8.** The results of model estimation.

| Variable (AQ) | Coef | Std. Err. | z | Prob |
|---|---|---|---|---|
| AuditA | 0.162 | 0.020 | 8.04 | 0.000 *** |
| Atenure | 0.013 | 0.017 | 0.75 | 0.454 |
| Achange | −0.044 | 0.079 | −0.56 | 0.575 |
| BIG1 | −0.058 | 0.062 | −0.95 | 0.341 |
| SIZE | −0.032 | 0.028 | −1.13 | 0.257 |
| ROE | 0.125 | 0.087 | 1.44 | 0.150 |
| ROA | −0.641 | 0.149 | −4.29 | 0.000 *** |
| Growthsales | 0.067 | 0.039 | 1.69 | 0.093 * |
| REST | 0.104 | 0.088 | 1.18 | 0.238 |
| Age | −0.002 | 0.002 | −0.87 | 0.388 |
| _con | 3.208 | 0.301 | 10.64 | 0.000 *** |
| Obs | 264 | | | |
| R-SQ | 0.2950 | | | |
| R-SQ2 | 0.2671 | | | |
| Prob Model | $F(10,253) = 10.59$ | | | |
| | Prob > F = 0.0000 *** | | | |

*** Significance level at 99% and * significance level at 90%.

**Table 9.** The results of model estimation.

| Variable (AQ) | Coef | Std. Err. | z | Prob |
|---|---|---|---|---|
| AQ L1. | −0.405 | 0.012 | −34.20 | 0.000 *** |
| AuditA | 0.172 | 0.032 | 5.31 | 0.000 *** |
| Atenure | −0.023 | 0.006 | −3.98 | 0.000 *** |
| Achange | −0.075 | 0.014 | −5.23 | 0.000 *** |
| BIG1 | −1.613 | 0.355 | −4.54 | 0.000 *** |
| SIZE | 0.042 | 0.015 | 2.85 | 0.004 *** |
| ROE | −0.050 | 0.032 | −1.59 | 0.113 |
| ROA | 0.005 | 0.046 | 0.11 | 0.912 |
| Growthsales | 0.060 | 0.013 | 4.44 | 0.000 *** |
| REST | 0.010 | 0.015 | 0.69 | 0.489 |
| Age | 0.032 | 0.005 | 5.75 | 0.000 *** |
| _con | 3.554 | 0.692 | 5.13 | 0.000 *** |
| Obs | 264 | | | |
| Prob Model | Wald chi2 (11) = 13,084.61 | | | |
| | Prob > chi2 = 0.0000 *** | | | |

*** Significance level at 99%.

### 4.7.3. The Results of the Research Model Estimation Based on the Combined Effects (BE) Method

To ensure that the research results are robust, we tested the research model with different methods to determine whether the results aligned with the main method's results. According to Table 10, based on the combined effects (BE) method, there is a positive and significant relationship between audit adjustments and quality because its *p*-value is 0.009, which is less than the significance level of 0.05, and its coefficients also are positive number 0.146. Therefore, it indicates that H0, which states a positive and significant relationship between audit adjustments and quality, is completely consistent with the main research method's results and is confirmed.

**Table 10.** The results of model estimation.

| Variable (AQ) | Coef | Std. Err. | z | Prob |
|---|---|---|---|---|
| AuditA | 0.146 | 0.051 | 2.85 | 0.009 *** |
| Atenure | −0.014 | 0.092 | −0.16 | 0.877 |
| Achange | −0.830 | 0.536 | −1.55 | 0.134 |
| BIG1 | 0.005 | 0.151 | 0.04 | 0.972 |
| SIZE | −0.356 | 0.160 | −2.23 | 0.036 ** |
| ROE | 0.532 | 0.358 | 1.49 | 0.150 |
| ROA | −1.017 | 0.352 | −2.89 | 0.008 *** |
| Growthsales | 0.192 | 0.171 | 1.12 | 0.274 |
| REST | 0.915 | 0.578 | 1.58 | 0.127 |
| Age | −0.006 | 0.004 | −1.39 | 0.179 |
| _con | 3.392 | 0.719 | 4.71 | 0.000 *** |
| Obs | 264 | | | |
| R-SQ | 0.1363 | | | |
| R-SQ2 | 0.0039 | | | |
| Prob Model | $F_{(10,24)} = 5.000$ | | | |
| | $Prob > F = 0.0006$ *** | | | |

*** Significance level at 99% and ** Significance level at 95%.

### 4.7.4. The Results of the Model Estimation Based on the Robust Regression Method

To ensure that the research results are robust, we tested the research model with different methods to determine whether the results aligned with the main method's results. According to Table 11, based on the stable or robust regression method, there is a positive and significant relationship between the audit adjustments and quality because its *p*-value is 0.000, which is less than the significance level of 0.05 and its coefficients also are positive number 0.197. Therefore, it indicates that H0 states a positive and significant relationship between audit adjustments and quality, which is consistent with the research method's results and confirms that.

**Table 11.** The results of model estimation.

| Variable (AQ) | Coef | Std. Err. | z | Prob |
|---|---|---|---|---|
| AuditA | 0.197 | 0.022 | 9.14 | 0.000 *** |
| Atenure | 0.015 | 0.018 | 0.81 | 0.418 |
| Achange | −0.063 | 0.084 | −0.76 | 0.451 |
| BIG1 | −0.022 | 0.066 | −0.34 | 0.732 |
| SIZE | −0.024 | 0.030 | −0.78 | 0.435 |
| ROE | 0.128 | 0.093 | 1.37 | 0.172 |
| ROA | −0.601 | 0.0159 | −3.76 | 0.000 *** |
| Growthsales | 0.080 | 0.043 | 1.88 | 0.061 * |
| REST | 0.119 | 0.094 | 1.27 | 0.206 |
| Age | −0.001 | 0.002 | −0.61 | 0.543 |
| _con | 3.709 | 0.322 | 11.51 | 0.000 *** |
| Obs | 264 | | | |
| Prob Model | F(10,253) = 11.99 | | | |
| | Prob > F = 0.0000 *** | | | |

*** Significance level at 99% and * significance level at 90%.

## 5. Discussion

The present study examined the relationship between audit adjustments and the quality of auditing services among the Iraqi listed companies. The research results showed a significant and positive relationship between audit adjustments and audit quality. This means that the greater the audit adjustments, the higher the audit quality. Such a finding might be explained by the monitoring role of auditors, in which the professional auditing services may motivate client firms to practice lower manipulation in accounting figures. For instance, it is expected that firms that experienced greater audit adjustment are less likely to show earning management, earning smoothing, and generally creative accounting. In this sense, Lennox et al. (2018) and Greenwood and Zhan (2019) stated that audit adjustments and financial reporting quality are closely correlated. To be more precise, the audit adjustments (auditor's proposed corrections) can affect the accruals quality model error components and reduce its average. Therefore, it seems that the proposed corrections by the auditor improve the quality of accruals (which in this study is a measure of the audit quality), which is in line with the obtained results of the research of Lennox et al. (2016, 2018). According to the research results, to improve the earning quality and the audit quality and more confidence of shareholders to respect their rights, auditors suggested that more and more address the necessary corrections during the year-end audit.

## 6. Conclusions

The findings of this paper suggest several contributions for equity owners and board members, as the decision makers for appointing auditors, as well as audit firms, as the professional bodies in the financial markets. Equity owners and board members might benefit from the findings of this investigation by appointing audit firms, which are well-known for their greater amount of audit adjustment among the market practitioners, because these professionals are more likely to improve the transparency and accuracy of provided financial statements, which are predicted to be a guideline for their decision-making process. Additionally, audit firms may improve their provided services through implementing additional reviews and analyses to come up with further audit adjustments, which might be translated into a favorable benchmark for high quality audit services by the market practitioners, which in turn may increase their market share and fees (Daemigah 2020).

Similar to other investigations, the authors of this paper also suffered from some limitations while conducting the study. The main limitation comes from the limited number of listed companies on the Iraq stock exchange; a greater number in this regard may increase the validity of the findings. Moreover, the limited empirical studies in the related literature have also precluded the authors from having a wider range of academic backgrounds for further comparisons and discussions.

According to the findings of this paper, we propose to prospective researchers to investigate the impact of audit adjustment on the expansion of the audit market and audit fees as well. As documented in the literature, greater audit quality might be a mechanism to increase the audit market concentration and audit fees of auditing companies. Thus, it is highly probable that auditors presenting further audit adjustments are willing to have a greater share in the market. Such an investigation may significantly contribute to the audit literature.

**Author Contributions:** Conceptualization, M.S. and M.I.J. methodology, M.S. software, S.O. validation, M.S. and M.I.J. formal analysis, A.M.A.A. investigation, M.S. and M.I.J. resources S.O. data curation, A.M.A.A. writing—original draft preparation M.I.J. writing—review and editing, S.O. visualization, M.I.J. supervision M.I.J. project administration, A.M.A.A. funding acquisition, S.O. All authors have read and agreed to the published version of the manuscript.

**Funding:** This research received no external funding.

**Institutional Review Board Statement:** Not applicable.

**Informed Consent Statement:** Not applicable.

**Data Availability Statement:** Not applicable.

**Conflicts of Interest:** The authors declare no conflict of interest.

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
