# Peer review of "The Relationship between Audit Adjustments and Audit Quality in Iraq"

_jrfm, doi:10.3390/jrfm15080330_

Round 1

Reviewer 1 Report

Please clearly indicate the research gap, which is the starting point for formulating research questions. There is extensive literature on various aspects of auditing, including its weaknesses. For example, please consider indicating whether auditors identify weak signals or whether and to what extent forensic auditing applies to your studies. Part of the article entitled the discussion and conclusions are minimal. Please expand it. Please indicate the limitations of the research and the directions of future research. It may help readers to understand the research better.

Author Response

Reviewer 1
Yes Can be improved Must be improved Not applicable
Does the introduction provide sufficient background and include all relevant references? ( ) ( ) (x) ( )
Are all the cited references relevant to the research? ( ) (x) ( ) ( )
Is the research design appropriate? (x) ( ) ( ) ( )
Are the methods adequately described? (x) ( ) ( ) ( )
Are the results clearly presented? (x) ( ) ( ) ( )
Are the conclusions supported by the results? ( ) ( ) (x) ( )
Comments and Suggestions for Authors
Please clearly indicate the research gap, which is the starting point for formulating research questions. There is extensive literature on various aspects of auditing, including its weaknesses. For example, please consider indicating whether auditors identify weak signals or whether and to what extent forensic auditing applies to your studies. Part of the article entitled the discussion and conclusions are minimal. Please expand it. Please indicate the limitations of the research and the directions of future research. It may help readers to understand the research better.

Response: some reasons are presented to justify the importance of conducting the current paper at the beginning of the introduction section. In other words, the flow of information presenting the academic gap regarding the paper's topic is provided. Moreover, forensic auditing in case of audit adjustment is conducted by prior literature (Lennox and Wu, 2022), resulting in lower novelty for further investigation, precluding the authors from exploring such an issue. Such an issue is mentioned in the introduction section. Finally, the discussion and conclusion sections are separated, both of which are improved by providing contributions, limitations and suggestions for future researchers. 

Reviewer 2 Report

Thank you for the opportunity to review this interesting article. After reading it I found the following aspects related to:

1. Abstract. The authors clearly state the main objective of their research, the results obtained and the conclusions.

2. Introduction. This section begins with the main objective of the research and makes the transition to the main causes that determined the investigation of the authors.

3. Literature review. After the completion of the first subsection 2.1. the working hypothesis of the authors is not specified, but only in subsection 2.2. I suggest the authors to insert this working hypothesis. ”Now, according to the above facts, the development of the hypotheses of the present research is as follows:” So where is it ???

4. Research methodology. This section is presented more accurately and I suggest the authors use a much clearer description so that all readers understand what is written. There is no mention of the data collection site or software used for data processing. I suggest the authors to do this! The description of the components of model (1) on page 7 should be taken directly on page 6 below the calculation relation!

5. Results. The authors analyze the data through tables created automatically in the data processing program but the explanations are far too precarious! I suggest the authors to clarify them!

6. Discussions and conclusions. I suggest that this section be divided into: Discussions and separate Conclusions. Not too many research findings are specified here, the importance of research and its implications in the field are not clearly highlighted, nor are the general conclusions or future research directions or the limits of the research emphasized! I suggest the authors to clarify the above!

Author Response

Reviewer 2
Yes Can be improved Must be improved Not applicable
Does the introduction provide sufficient background and include all relevant references? ( ) (x) ( ) ( )
Are all the cited references relevant to the research? ( ) (x) ( ) ( )
Is the research design appropriate? ( ) (x) ( ) ( )
Are the methods adequately described? ( ) (x) ( ) ( )
Are the results clearly presented? ( ) (x) ( ) ( )
Are the conclusions supported by the results? ( ) (x) ( ) ( )
Comments and Suggestions for Authors
Thank you for the opportunity to review this interesting article. After reading it I found the following aspects related to:
1. Abstract. The authors clearly state the main objective of their research, the results obtained and the conclusions.
Response: the abstract section is improved and further details are added to present a clearer overview of the entire paper.
2. Introduction. This section begins with the main objective of the research and makes the transition to the main causes that determined the investigation of the authors.
Response: some reasons are presented to justify the importance of conducting the current paper at the beginning of the introduction section. In other words, the flow of information presenting the academic gap regarding the topic of the paper is provided.
3. Literature review. After the completion of the first subsection 2.1. the working hypothesis of the authors is not specified, but only in subsection 2.2. I suggest the authors to insert this working hypothesis. ”Now, according to the above facts, the development of the hypotheses of the present research is as follows:” So where is it ???
Response: The author's intention for the sentence “Now, according to the above facts, the development of the hypotheses of the present research is as follows:” was the next section of the paper which is “2.2.. However, to make it more apparent, the sentence is updated.
4. Research methodology. This section is presented more accurately and I suggest the authors use a much clearer description so that all readers understand what is written. There is no mention of the data collection site or software used for data processing. I suggest the authors to do this! The description of the components of model (1) on page 7 should be taken directly on page 6 below the calculation relation!
Response: additional details about the source of collected data (the official website of Iraqi stock exchange) is mentioned in a footnote under the section “3.1. The method and data collection” and the applied tool for data are analyzing is presented under the “3.2. The method of data analysis” section. Furthermore, the variable definitions are presented directly on page 6 below the calculation relation.
5. Results. The authors analyze the data through tables created automatically in the data processing program but the explanations are far too precarious! I suggest the authors to clarify them!
Response: results are elaborated more and some supportive evidence is provided as well.
6. Discussions and conclusions. I suggest that this section be divided into: Discussions and separate Conclusions. Not too many research findings are specified here, the importance of research and its implications in the field are not clearly highlighted, nor are the general conclusions or future research directions or the limits of the research emphasized! I suggest the authors to clarify the above!
Response: the discussion and conclusion sections are separated, both of which are improved by providing contributions, limitations and suggestions for future researchers.

Reviewer 3 Report

The article received for review is an interesting one. However, there are some aspects that I found and are related to:

Introduction. This section specifies the main objective of the research but does not specify the lack of literature or the reason for differentiating between the present study and the previous ones. Please clarify this issue.

Specialty literature. The first subsection presents some interesting aspects related to the quality of the audit and the findings of the specialists, but it does not present the working hypothesis H0 (page 4). The second subsection presents the link between audit adjustments and audit quality. The hypothesis launched is a correct one. Overall, this section should be improved with other more recent studies from 2020 to the present. Please do this.

Research methodology. It is a precarious one, the authors not specifying too many concrete aspects related to the investigations carried out. Based on a research model, the authors try to demonstrate its viability in the launched context.

Analyze the results. The testing of the missing H0 hypothesis is specified. To understand and validate the results obtained we should know which hypothesis H0! Until it is specified, I cannot comment on the validity of the processed data! Please ask the authors to fill in the missing information!

Discussions and conclusions. This subsection must be separated. The authors must present in the Discussions section all aspects related to the analysis of the results and their interpretation and in the Conclusions section to present the general conclusions obtained as a result of the case study, its limits and future research possibilities related to this topic. I ask the authors to achieve these things!

Author Response

Reviewer 3
Yes Can be improved Must be improved Not applicable
Does the introduction provide sufficient background and include all relevant references? ( ) (x) ( ) ( )
Are all the cited references relevant to the research? ( ) (x) ( ) ( )
Is the research design appropriate? ( ) (x) ( ) ( )
Are the methods adequately described? ( ) (x) ( ) ( )
Are the results clearly presented? ( ) (x) ( ) ( )
Are the conclusions supported by the results? ( ) (x) ( ) ( )
Comments and Suggestions for Authors
The article received for review is an interesting one. However, there are some aspects that I found and are related to:
Introduction. This section specifies the main objective of the research but does not specify the lack of literature or the reason for differentiating between the present study and the previous ones. Please clarify this issue.
Response: some reasons are presented to justify the importance of conducting the current paper at the beginning of the introduction section. In other words, the flow of information presenting the academic gap regarding the topic of the paper is provided.
Specialty literature. The first subsection presents some interesting aspects related to the quality of the audit and the findings of the specialists, but it does not present the working hypothesis H0 (page 4). The second subsection presents the link between audit adjustments and audit quality. The hypothesis launched is a correct one. Overall, this section should be improved with other more recent studies from 2020 to the present. Please do this.
Response: the author's intention for the sentence “Now, according to the above facts, the development of the hypotheses of the present research is as follows:” was the next section of the paper, "2.2.. However, to make it more apparent, the sentence is updated. Several studies published from 2020 to 2022 are added to update the literature section.

Research methodology. It is a precarious one, the authors not specifying too many concrete aspects related to the investigations carried out. Based on a research model, the authors try to demonstrate its viability in the launched context.
Response: To improve the model's validity and obtain results, the authors have applied some robustness tests.
Analyze the results. The testing of the missing H0 hypothesis is specified. To understand and validate the results obtained we should know which hypothesis H0! Until it is specified, I cannot comment on the validity of the processed data! Please ask the authors to fill in the missing information!
Response: the “H0 hypothesis” was a writing error made by the authors, which is corrected in the new version the term is replaced with “the hypothesis” because the paper has only one hypothesis. The results are more discussed and updated as well.
Discussions and conclusions. This subsection must be separated. The authors must present in the Discussions section all aspects related to the analysis of the results and their interpretation and in the Conclusions section to present the general conclusions obtained as a result of the case study, its limits and future research possibilities related to this topic. I ask the authors to achieve these things!
Response: the discussion and conclusion sections are separated, both of which are improved by providing contributions, limitations and suggestions for future researchers.

Round 2

Reviewer 1 Report

Please expand your literature research. Audit quality is not only determined by the quality of the process. It is also the quality of the methods used. Please use the following articles: Implementing a sustainable model for anti-money laundering in the United Nations development goals;  The supreme audit institutions readiness to uncertainty; Forensic auditing and weak signals: a cognitive approach and practical tips; or similar articles showing the problems from a multifaceted perspective. 

Please make sure that the proofreading is correct. For example, the subordinate phrase "to achieve the objectives" does not appear to modify the subject "the multivariate regression model". Consider whether to rewrite the sentence to avoid a dangling modifier. 

Reviewer 2 Report

I thank the authors for making the suggestions.

Reviewer 3 Report

I want to thank the authors for taking into account all the suggestions I indicated. I am satisfied with the added ones.